



# Understanding the model representation of clouds based on visible and infrared satellite observations

Stefan Geiss[1], Leonhard Scheck[1,2], Alberto de Lozar[2], and Martin Weissmann[3]

[1]Hans-Ertel Centre for Weather Research, Ludwig-Maximilians-Universität, Munich, Germany
[2]Deutscher Wetterdienst, Offenbach, Germany
[3]Institut für Meteorologie und Geophysik, Universität Wien, Vienna, Austria

**Correspondence:** S. Geiss (s.geiss@physik.uni-muenchen.de)

**Abstract.** Satellite observations provide a wealth of information on atmospheric clouds and cover almost every region of the globe with high spatial resolution. The measured radiances constitute a valuable data set for evaluating and improving clouds and radiation representation in climate and numerical weather prediction (NWP) models. An accurate, bias-free representation of clouds and radiation is crucial for data assimilation and the increasingly important solar photovoltaic (PV) power production

prediction. The present study demonstrates that visible (VIS) and infrared (IR) Meteosat SEVIRI observations contain valuable and complementary cloud information for these purposes.

We analyse systematic deviations between satellite observations and convection-permitting, semi-free ICON-D2 hindcast simulations for a 30-day period with strong convection. Both visible and infrared satellite observations reveal significant deviations between the observations and model equivalents. The combination of infrared brightness temperature and visible solar

reflectance allowed to attribute individual deviations to specific model shortcomings. Furthermore, we investigate the sensitivity of model-derived VIS and IR observation equivalents to modified model and visible forward operator settings to identify dominant error sources. The results reveal that model assumptions on subgrid-scale water clouds are the primary source of systematic deviations in the visible spectrum. Visible observations are, therefore, well-suited to advance this essential model assumption. The visible forward operator uncertainty is lower than uncertainties introduced by model parameter assumptions

by one order of magnitude. In contrast, infrared satellite observations are very sensitive to ice cloud model assumptions. Finally, we show a strong negative correlation between VIS solar reflectance and global horizontal irradiance. This implies that improvements in VIS satellite reflectance prediction will coincide with improvements in the prediction of surface irradiation and PV power production.

# 1   Introduction

Satellite observations provide an indispensable data source with a high spatial and temporal resolution for evaluating model clouds, particularly when conventional cloud observations are sparse or missing. Especially satellite observations in the visible





range and infrared channels in atmospheric windows with low absorption by atmospheric trace gases are very sensitive to the presence of clouds. Approximately, all radiation reaching the satellite instrument originates from liquid (e.g. water clouds)

and solid particles (e.g. ice clouds, aerosols) in the atmosphere or the earth's surface. Hence, the radiation flux is determined by the atmospheric state and surface properties. While reflection of solar radiation dominates in the visible range, thermal emission is the primary source of radiation in the infrared. The statistical comparison of model cloud properties with satellite observations is an important tool for evaluating and improving the representation of clouds in numerical models (Roebeling et al., 2006). Prerequisite for a fair comparison of model fields with observations is an appropriate transform, either from model

to observation space or from observation to model space. The latter approach, where observations are transformed to model variables has potential for evaluating cloud properties like cloud fraction, cloud optical depth, and cloud top height. However, retrieving these quantities from satellite-measured radiances introduces errors with magnitudes of up to 40-80%. These errors make it challenging to evaluate clouds produced by an NWP model (Jonkheid et al., 2012). The other option is to assess observed and simulated radiances directly in observation space. The errors of forward operators that calculate observation

equivalents from the model state are usually significantly smaller than that of the retrieval procedure. Furthermore, it is easier to quantify the embedded uncertainties (Reitter et al., 2011), and we, therefore, pursue the comparison in observation space.

Besides model evaluation, satellite observations are an important source of information for data assimilation, nowcasting and forecast verification. For these purposes, forward operators that calculate model equivalents of the observations have to be both computationally efficient and accurate. In the infrared, the radiative transfer package RTTOV (Saunders et al., 2018)

fulfils these criteria and is operationally used by many weather centres (e.g. ECMWF). Related uncertainties were examined by several authors (e.g. Senf and Deneke, 2017; Saunders et al., 2017, 2018). The newly developed fast and precise forward operator VISOP is applied for visible channels, which is based on the 1D radiative transfer method MFASIS Scheck et al. (2016). Clouds must be adequately analysed, since they affect the model's energy balance and indicate locations of possible convective initiation (Mecikalski et al., 2013). By applying forward operators to model state, clouds can be easily examined based on

the comparison to observations. Furthermore, the minimisation of systematic errors is a prerequisite for data assimilation, and the observations can only be assimilated if observed and simulated clouds exhibit a similar climatology. Unfortunately, this is not necessarily the case for cloud-related quantities in current NWP systems. Therefore, understanding and mitigating these systematic deviations will be an essential ingredient for the operational assimilation of such observations. Some studies already showed the benefit of assimilating cloud-related quantities in the infrared (e.g. Geer et al., 2018; Honda et al., 2018) and in the

visible (Scheck et al., 2020) in experimental setups or idealized experiments (Schröttle et al., 2020). The assimilation led to significant improvements in cloud-related quantities and dynamical variables as clouds are often associated with meteorologically sensitive areas (McNally, 2002). However, current convection-permitting operational NWP systems still do not assimilate cloud-affected satellite observation (Gustafsson et al., 2018), which is to a large extent due to issues with systematic deviations. Improving the model representation of clouds is also essential, given the rising share of renewable energy in the world's total

electricity supply. While solar photovoltaic (PV) power production is one of the fastest-growing forms of renewable energy, with an increase of 22% in 2019 (IEA, 2020), it will soon become challenging to integrate PV power into the electricity grid given its strong weather-related fluctuations. The accurate prediction of these fluctuations based on NWP models is important





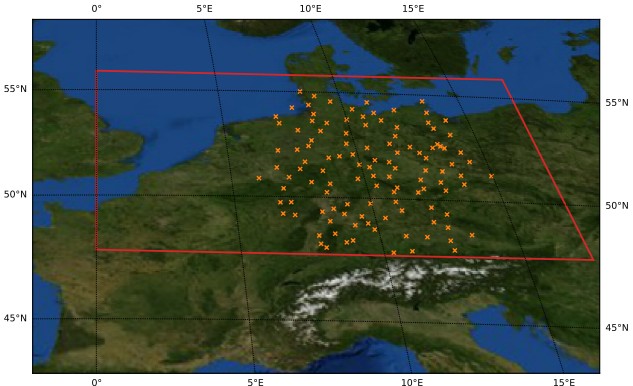

**Figure 1.** ICON-D2 and reduced verification domain (red box) from MSG's point of view. Orange x indicate the 122 pyranometer stations measuring global radiation.

in terms of network safety and the efficient deployment of alternative power sources. More specifically, solar irradiance fluctuations determine 90 % of the output power fluctuations of a photovoltaic (PV) power plant. Solar irradiance fluctuations, in turn, are mainly determined by atmospheric clouds (Zack, 2011). According to Köhler et al. (2017), the main shortcomings of NWP in this context is the prediction of low stratus and fog, the spatial and temporal resolution of convection, shallow cumulus and Saharan dust outbreaks. Kurzrock et al. (2018) also demonstrated that clouds and in particular, the representation of low stratus in the model dominate the uncertainty of PV power production. However, a better representation of these meteorological features is challenging due to the subgrid-scale nature of clouds and the lack of aerosols information.

In this paper, we evaluate and analyse shortcomings of the representation of clouds in the preoperational convection-permitting ICON-D2 model of Deutscher Wetterdienst based on the comparison to satellite observations for a 30-day highly convective period. For the generation of synthetic MSG SEVIRI satellite observations from the model state, we applied the forward operators VISOP and RTTOV. Through this, systematic deviations became apparent in solar reflectance (VIS006) and brightness temperature (IR108) cloud distributions. Moreover, we combined these two channels and computed 2D PDFs, revealing additional shortcomings as both channels contain complementary information. To better understand the reasons for these deviations, we perturbed cloud-related model parameters to examine their effects on synthetic VIS006 and IR108 observations. For the visible forward operator VISOP, we investigated operator uncertainties related to 3D effects, overlap assumptions, aerosols and ice habits. As RTTOV is a well-established RT package and other studies already investigated its uncertainties, we did not assess operator sensitivities for the infrared channels. Finally, we determined the dependence of global horizontal irradiance (GHI) on VIS006 and IR108, respectively.

The remainder of the paper is structured as follows: The experimental setup is presented in section 2. Two selected days with clouds on different levels are analysed in section 3 to introduce satellite observations and their characteristics. This is followed by a discussion of the cloud climatology and associated systematic deviations. Then, we investigate the dependence between surface radiation and the utilised satellite channels. In section 4, we assess the sensitivity of synthetic satellite images to



model and visible operator settings. For solar reflectances, forward operator uncertainties and model sensitivity are compared. Conclusions are provided in section 5.

## 2   Experimental setup

### 2.1   Model setup and sensitivity experiments

To evaluate the cloud climatology during this period, we use the pre-operational convection-permitting ICON-D2 (ICOsahedral
Non-hydrostatic; (Zängl et al., 2015)) model configuration with prescribed lateral boundary conditions (BCs) and a one-way nesting. ICON-D2 will replace the operational COSMO-D2 model (Baldauf et al., 2018). Simulations over Germany with a horizontal grid spacing of 2.1 km and 65 vertical levels are initialised once at 26 May 2016 00 UTC from downscaled ICON-EU analysis initial conditions. ICON-EU analysis BCs drive this semi-free simulation with an hourly update and a forecast horizon of 30 days. The simulation period and domain size are sufficiently large for the atmospheric model to develop its
own cloud climatology without perturbations from data assimilation or nudging. In our reference simulation, the operational single-moment bulk microphysical parameterisation accounting for cloud water, cloud ice, snow, rain and graupel is used (Lin et al., 1983; Reinhardt and Seifert, 2006).

The reference pre-oprational model configuration has been reached through extensive tuning of many parameters whose values are uncertain. Since many of these parameterisations are related to clouds, it would be very beneficial if such parameter-
isations could be further constrained by satellite observations. For this reason we examine the sensitivity of solar reflectances and infrared brightness temperatures to variations in cloud-related parameterisations. We performed six simulations in which cloud-related parameterisations were modified within their range of uncertainty, i.e. using perturbed values that are physically plausible. The objective was to determine which parameterisations produce a change in the synthetic-satellite signal beyond the uncertainties of the forward operator, and therefore can be improved by using satellite observations. For this purpose, we
modified the following four parameterisations:

1. The cloud-concentration number in ICON is used to calculate the cloud optical properties and the onset of precipitation. ICON employs the parameterisation of Segal and Khain (2006), which determines how many droplets are in a cloud depending on an aerosol number concentration derived from the climatology and on an updraft velocity at nucleation. The determination of the updraft velocity in a 2 km resolution model is not straightforward, because updrafts are under
resolved. ICON assumes a constant updraft velocity, which serves as a control parameter: the number of nucleated droplets increases with the updraft velocity.

2. The turbulent subgrid-scale cloud parameterisation determines the cloud cover due to the unresolved variability in the model. We focus on the parameterisation of liquid clouds because those are more abundant than ice clouds in the summer period chosen for the experiments. Not including limiters and other correction factors which will not be discussed here,





the cloud cover in ICON is a function of the absolute humidity:

$$cc = \left( \frac{q_v + q_c + A\Delta q - q_{sat}}{B\Delta q} \right)^2, \tag{1}$$

where $q_v$ is the vapor water content, $q_c$ is the cloud-water content, $\Delta q$ is a parameter that quantifies the variance of water inside the grid box (mostly determined by the turbulent scheme) and $B$ and $A$ are two tunable parameters. In the pre-operational configuration it is set $A = 3.5$ and $B = 1 + A = 4.5$. The parameter $A$ (called asymmetry factor, which

should not be confused with the factor $g$ in radiation) determines the number of subgrid clouds. This is a common tuning parameter when changing the model configuration. For example, it is expected that the model requires less subgrid clouds as grid spacing is reduced and more clouds are resolved.

3. The shallow-convection parameterisation of Bechtold et al. (2014) predicts unresolved shallow convection in the model and also contributes to subgrid clouds. The model limits the parameterisation to clouds that are sufficiently thin, so that

thicker clouds have to be resolved by the model. The thickness of the thickest non-resolved cloud is thus an uncertain parameter that limits the strength of the parameterisation.

4. The microphysical scheme describes the hydrometeors dynamics. We check the effect of using the two-moment parameterisation of Seifert and Beheng (2006), in which the number concentrations of different variables are treated as prognostic variables. This is a more complex scheme and can potentially simulate more realistic clouds, However, the

two-moment scheme has never been tuned like the operational one.

In order to investigate the sensitivity of satellite synthetic observations to these parameterisations we have evaluated seven simulations:

I Reference simulation with pre-operational model configuration.

II Decreased cloud-concentration number by increasing the updraft velocity at activation ( from $0.25\,\mathrm{m/s}$ to $1\,\mathrm{m/s}$). This

produces liquid clouds that are optically thicker as the number concentration of droplets increases roughly by a factor three.

III Modified distribution of turbulent subgrid liquid clouds. The idea is to produce less and thicker subgrid clouds in a way that the radiative balance of the model remains unchanged. This was achieved after a few trial and error experiments by using the parameters $A = 2.5$, $B = 0.21$.

IV Stronger shallow-convection parameterisation by doubling the thickness of the thickest unresolved cloud (from $2 \cdot 10^4$ to $4 \cdot 10^4$ hPa).

V Simulation with the two-moment scheme while all other parameterisations are equal to the operational configuration.

VI Two-moment scheme in which the subgrid-cloud parameterisation for ice clouds is switched off.

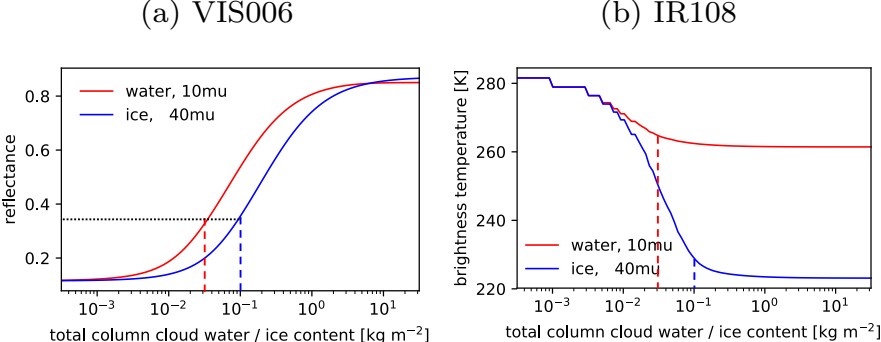

**Figure 2.** Water and ice cloud signals with different effective particle radii from VIS006 (a) and IR108 (b), computed using DISORT. Dashed lines indicate saturation in IR108 for water (red) and ice (blue) clouds. The albedo was set to 0.1, the solar zenith angle to $30°$, the satellite zenith angle to $60°$ and the scattering angle to $135°$.

VII Two-moment scheme with strongly reduced asymmetry factor for subgrid-liquid clouds ($A = 1.5$, $B = 2.5$) and no subgrid ice-clouds. This simulations was motivated because the two-moment scheme reflected too much radiation, and therefore we reduced the amount of subgrid clouds.

### 2.2 Satellite observations

The SEVIRI instrument onboard METEOSAT Second Generation has eight channels in the solar and thermal part of the atmospheric window (Schmetz et al., 2002). In the solar regime, radiances are dominated by scattering of photons from the sun to the satellite sensor, while emission of the earth's surface and cloud top is dominant in the thermal. In this paper, we use the visible $0.6\,\mu m$ channel (VIS006), which has the advantage that at this wavelength the surface albedo is usually relatively low (R<0.15) and thus also errors in the albedo are smaller than for the $0.8\,\mu m$ channel (VIS008) that would also be available from SEVIRI. Additionally, we use the $10.8\,\mu m$ thermal infrared window channel (IR108). At this wavelength, the signal is not strongly affected by gaseous absorption within the atmosphere and mainly determined by emission from the ground and clouds at all heights. For a better understanding and interpretation of our results, we discuss the sensitivity of the VIS006 and IR108 signals to total column cloud water (TCW) and ice (TCI), as shown in Fig. 2. The signals are computed using DISORT (DIScrete Ordinates Radiative Transfer; Stamnes et al. (2000)) for idealised scenes with a single-layer water cloud at the height of $4\,km$ or a single-layer ice cloud at the height of $10\,km$.

Both, solar reflectance and infrared brightness temperature strongly depend on TCW and TCI, but in different ranges: VIS006 is most sensitive to TCW/TCI-values in the range $[10^{-2}, 10^{0}]\,kg/m^2$. In comparison, IR108's sensitivity is limited to thinner clouds with values in the range $[10^{-2}, 10^{-1}]\,kg/m^2$, due to a fast saturation of the signal by the absorption of photons. Figure 2b implies that for a single-layer water cloud with TCW>$0.03\,kg/m^2$ or a single-layer ice cloud with TCI>$0.1\,kg/m^2$, only cloud top height and its corresponding temperature determines the observed BT. The IR signal can thus provide the cloud top





temperature but does not allow for retrieving the TCW/TCI. In contrast, the solar reflectance is only 0.35 at these threshold
values and can still provide information on the water/ice content up to TCW/TCI values of about $1\,\text{kg/m}^2$. As the flux of the
reflected radiation is anticorrelated with the flux of the transmitted radiation (which will be discussed more quantitatively in
Sect. 3.3), solar reflectances are sensitive to changes in cloud water/ice content in the range where such changes have also the
strongest impact on the incoming solar radiation at the surface.

These different and complementary sensitivities show that model evaluation with solar and thermal channels has the po-
tential to provide more information on the nature of the systematic errors and to possibly identify specific shortcomings that
would not be visible by only examining a single channel.

## 2.3    Satellite forward operators

To compute model equivalents for visible satellite images from the ICON model state, we employ the VISible satellite image
Forward OPerator (VISOP) that uses the fast 1D radiative transfer (RT) method MFASIS (Scheck et al., 2016). MFASIS is
based on a compressed lookup table (LUT), computed using the DISORT solver. It is possible to consider aerosols or different
kind of ice habits for the computation of the MFASIS LUT (results in section 4.2). VISOP takes the slant satellite viewing
angle into account (tilted independent column approximation; (Wapler and Mayer, 2008)) and accounts for the most important
3D RT effect by using the cloud top inclination correction (CTI) described in Scheck et al. (2018). The surface albedo values
required as input for MFASIS are taken from the RTTOV-BRDF Atlas (Vidot et al., 2018).

As we aim to achieve consistent assumptions in both the operator and the NWP model, we decided to use effective radii
from the model output for water clouds directly. This is based on the consideration that radiative transfer, micro-physics and
possibly operators should deal with the same optical properties.

However, some adjustments are required for the ice clouds, as will be motivated in the following. The micro-physics scheme
in the simulation predicts six hydrometeor categeories: cloud water, cloud ice and precipitating liquid water, snow, hail, and
graupel. Rain droplets, hail and graupel particles are assumed to be much larger than cloud droplets and cloud ice particles
in the model. Therefore, for the same mass they are also much less effective in scattering radiation and are thus neglected in
the forward operators. However, the distinction between snow and cloud ice particles in the model is rather artificial. Model
snow particles can be small enough to cause non-negligible scattering effects (see discussion in Hogan et al., 2001). Hence, as
a (first) approximation we construct a frozen phase whose total mass, $Q_i^{tot}$, is the sum of the diagnosed ice water content (grid
and subgrid-scale) and snow content (only grid scale available) and whose "effective effective radius",

$$R_{I,eff}^{tot} = \frac{Q_i^{tot}}{(Q_i^{DIA}/r_{i,eff} + Q_s/r_{s,eff})}, \tag{2}$$

is defined using the simulated effective radii of cloud ice $r_{i,eff}$ and snow $r_{s,eff}$. This approximation assumes that the optical
thickness of the frozen phase is equal to the sum of the optical thicknesses of the ice and snow phases. The approximation be-
comes exact in case of wavelengths much smaller than the hydrometors size (optical limit), and therefore it is quite appropriate
for visible channels.





In general, we use the diagnosed cloud water- and ice content including subgrid contributions as input for VISOP. If no subgrid-scale cloud is diagnosed in a particular grid box, then $Q_x^{DIA} = Q_x$, where x could be either water or ice.

An accurate calibration is a prerequisite for using satellite observations, but unfortunately the calibration of SEVIRI VIS006 is
uncertain. Meirink et al. (2013), for example, found a calibration offset of - 8 % for VIS006 during the years 2004 to 2008 by comparing MSG SEVIRI and MODIS (Moderate Resolution Imaging Spectroradiometer) Aqua observations. For our purpose, we use the approach to find a suitable calibration offset by minimising the average histogram difference between the observed and simulated solar reflectance distribution. Through this, we found an offset of -13 % between observations and our reference simulation.

To derive SEVIRI infrared brightness temperature from the model state, we use the efficient methods implemented in the RT-TOV 12.1 package (Saunders et al., 2018), which is used by many weather services. The spatial resolution of MSG SEVIRI VIS006 and IR108 observations is 3 km x 3 km at subsatellite point and reduces to 6 km x 3 km in the ICON-D2 domain, with a temporal resolution of 15 min.

For the evaluation, we applied both operators at the full model resolution and interpolated solar reflectances and brightness
temperature to observation space afterwards to avoid additional representativeness errors. (Marseille and Stoffelen, 2017).

## 2.4   Global horizontal irradiance observation and forward operator

Global horizontal irradiance (GHI) is the total amount of shortwave radiation and includes both direct normal irradiance (DNI) and diffuse horizontal irradiance (DHI). Deutscher Wetterdienst operates a dense network of GHI observations across Germany. GHI is an hourly average and is evaluated at 122 pyranometer stations (Fig. 1). The model's radiative transfer scheme RRTM
(Rapid Radiative Transfer Model) simulates DNI and DHI (Mlawer et al., 1997).

## 2.5   Verification metrics

A combination of metrics is applied to evaluate synthetic satellite imagery at 12 UTC with observations. The verification domain (red rectangle in Fig. 1) is smaller than the ICON-D2 domain to exclude nesting effects at the domain boundaries and signals from snow-covered surfaces in the Alps that exhibit reflectances similar to clouds. We show VIS006 and IR108
probability density functions (PDF) of our simulations $P^{sim}$ and calibrated observations $P^{obs}$, without coarsening or thinning. The number of bins N of the PDFs is 50, with R∈[0,1] and BT∈[200,300] K. From that, we define the cloudiness (C) as the fraction of pixels in which the solar reflectance is higher than a threshold value $R_c$ of 0.2. This value is an upper limit for the clear-sky reflectance in the considered verification domain (see discussion in Scheck et al. (2018)). Violin plots are used to visualize the daily bin-by-bin deviation of the PDF (deviation computed for each day d and bin n) from the reference
run and model/operator sensitivity experiments: $\epsilon_{n,d}^{hist} = P_{n,d}^{obs} - P_{n,d}^{sim}$. This allows for a consistent comparison of VISOP and model uncertainty, by examining the median deviation (the mean is always zero), the interquartile range (difference between 75th and 25th percentile) as a measure for variability and the range as the extent of deviations. We further analyze clouds by constructing contoured 2D probability density function (PDF) plots of brightness temperature and solar reflectance, comparable to contoured frequency by altitude diagrams (CFADs) of radar observations. We use the US. Standard Atmosphere 1962





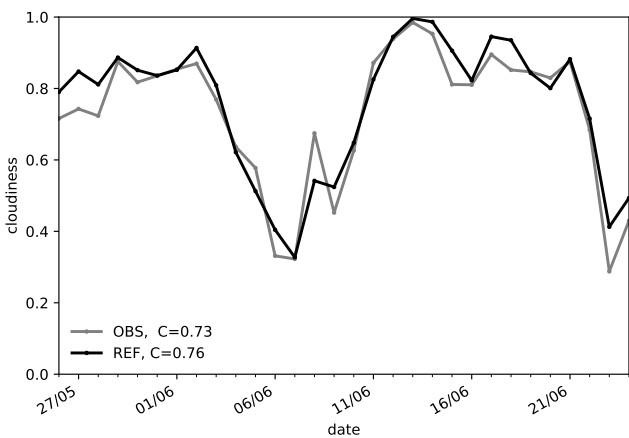

**Figure 3.** Time series of observed and simulated cloudiness at 12 UTC during period (26 May - 24 June 2016).

(Sissenwine et al., 1962) to classify brightness temperatures into three cloud categories (low, middle and high clouds) as defined in the International Cloud Atlas (Cohn, 2017). In the US Standard Atmosphere, the surface temperature is 288 K and the (wet) temperature lapse rate is 0.65 K/100 m, leading to temperature ranges of T > 275 K for the surface and low clouds, 275 K ≤ T ≤ 243 K for middle clouds and T < 243 K for high clouds.

## 2.6 Synoptic overview and cloudiness

A 30-day period from 26 May to 24 June 2016 is analysed, which is dominated by strong summer-time convection in Germany. In the beginning, large parts of Europe were affected by high-impact weather events over almost two weeks. Atmospheric blocking and interaction of low thermal stability and weak mid-tropospheric winds were the ingredients for this exceptional sequence of thunderstorms and related flash floods (Piper et al., 2016). Many authors have discussed these two weeks (see e.g. Necker et al. (2020); Bachmann et al. (2020); Keil et al. (2019); Necker et al. (2018); Zeng et al. (2018)). In the subsequent

weeks (10. - 24. June), the wind direction changed to south-westerly flow, advecting warm and humid air masses from the Atlantic and the Mediterranean to Germany and supporting cloud formation (Fig. 3). In general, the simulated cloudiness (defined in section 2.5) is predominantly overestimated, leading to a period mean observed and simulated cloudiness of 0.73 and 0.76, respectively. This convective period with high cloud cover at different levels is well suited to examine the cloud climatology and its sensitivity to cloud-related parameterisations.

## 3 Reference run


### 3.1 Selected cases

In this section, we exemplarily discuss two days of the period to illustrate the methodology of evaluating clouds using visible and infrared satellite channels. On the first one (29 May), deep convection and severe thunderstorms occurred leading to a





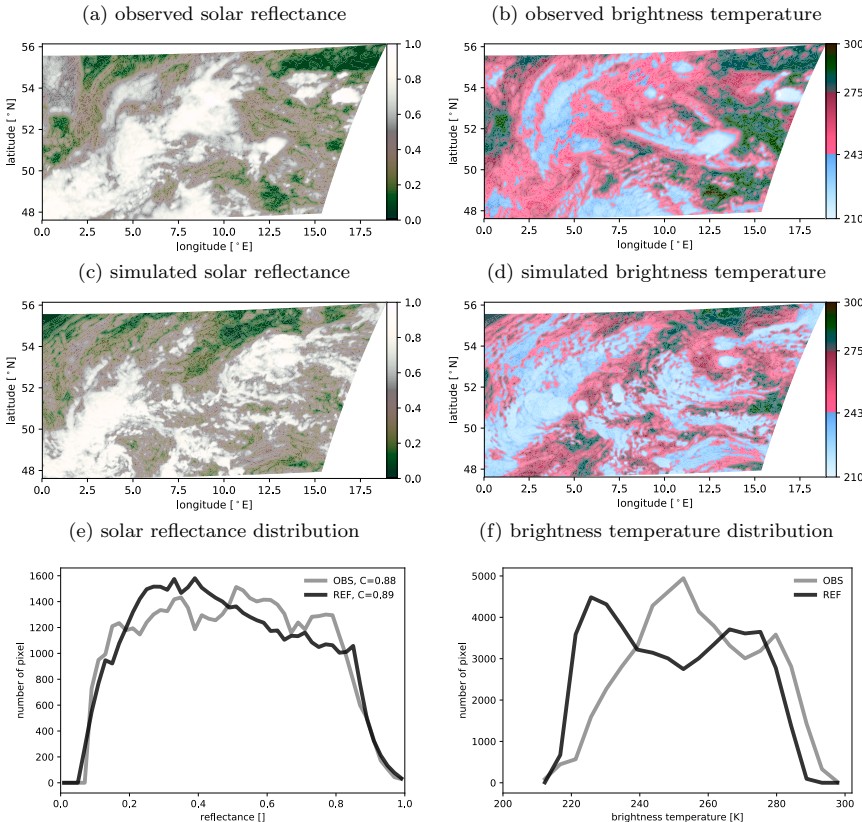

**Figure 4.** (Regional) distribution of solar reflectances (top) and brightness temperatures (bottom) and their corresponding distribution for 29 May at 12 UTC (EUMETSAT).

flash flood that caused severe damage in Braunsbach, a small town in the south-western part of Germany. The second one (02
June) was dominated by low-level clouds. According to Piper et al. (2016), warm, moist and unstable air masses characterized both days. However, large-scale ascent dominated on 29 May and subsidence on 02 June. Figure 4 shows the VIS006 and IR108 satellite images, together with the corresponding distributions of solar reflectance and brightness temperatures on 29 May 2016. The VIS006 satellite image (Fig. 4a & 4b) shows the early stage of a cyclogenesis over Germany, characterized by a prominent vortex structure, in both the observation and model simulation. However, the feature is shifted to the south-west
in the simulation. The relatively high cloudiness of 88 % in the observation and 89 % in the simulation leads to a relatively uniform distribution of observed solar reflectances (Fig. 4e). Overall, the agreement between observed and simulated visible histograms is relatively good given that the model is forced towards the current weather only through the boundary conditions. The vortex structure of the cyclogenesis is also apparent in the IR108 observation (Fig. 4b), but the simulation shows clear systematic errors. In the simulation, the cloud pattern is dominated by relatively high ice clouds (Fig. 4d), which are less fre-



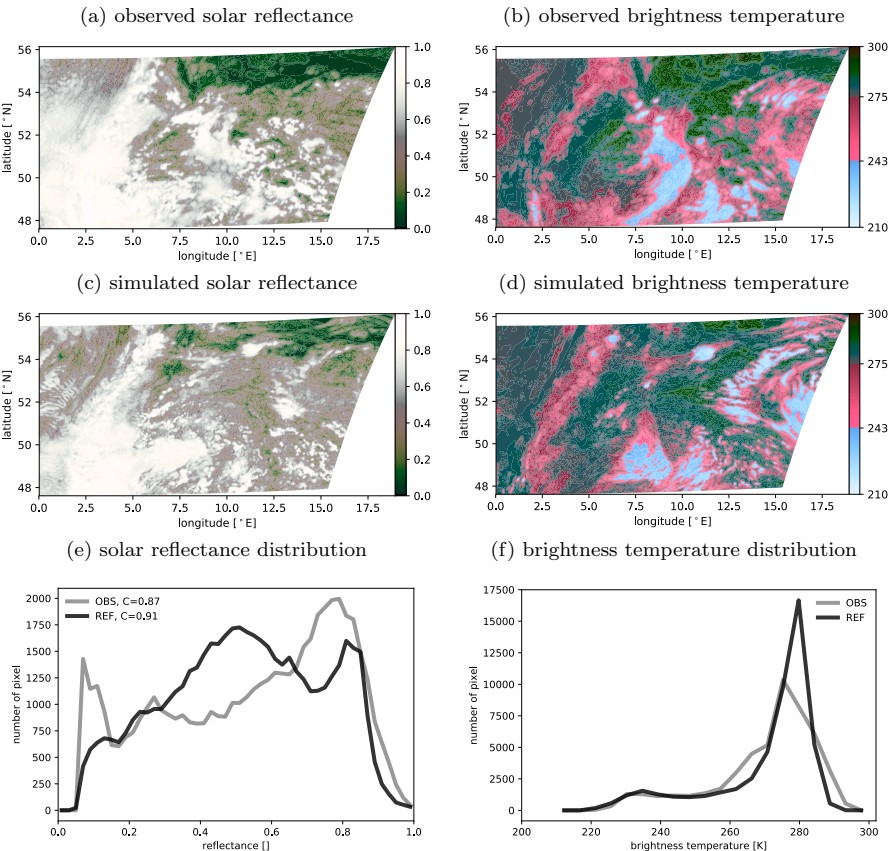

**Figure 5.** As in Fig. 4 but for 02 June 2016 (EUMETSAT).

quent in the observations. The histogram confirms this picture: The signal of high clouds is overestimated in the simulations, whereas the signal of medium clouds is underestimated by 40 %.

On 02 June 2016, boundary layer clouds dominated in both the observation and simulation (Fig. 5b&d). Additionally, superimposed ice clouds are observed in some regions. The simulated IR108 distribution fits the observed one relatively well on this day (Fig. 5f). In the visible satellite image (Fig. 5a&c), a high cloudiness is apparent, with 87 % in the observation and 91 % in the simulation. Different to 29 May, however, the distribution (Fig. 5e) reveals an overestimation of medium-thick clouds, together with an underestimation of thick clouds (R>0.6).

The examples discussed above show that the examination of a single channel (VIS or IR) can lead to opposite conclusions with respect to forecast quality. The agreement of the histograms for 29 May is good in the visible range but not in the IR. The opposite is observed for the 02 June. This shows that both channels provide complementary information. In the following, we show that further information can be obtained by using the combined information of both channels in 2D PDF plots of brightness temperature and reflectance. We have already discussed how the IR histogram shows an overestimation of high



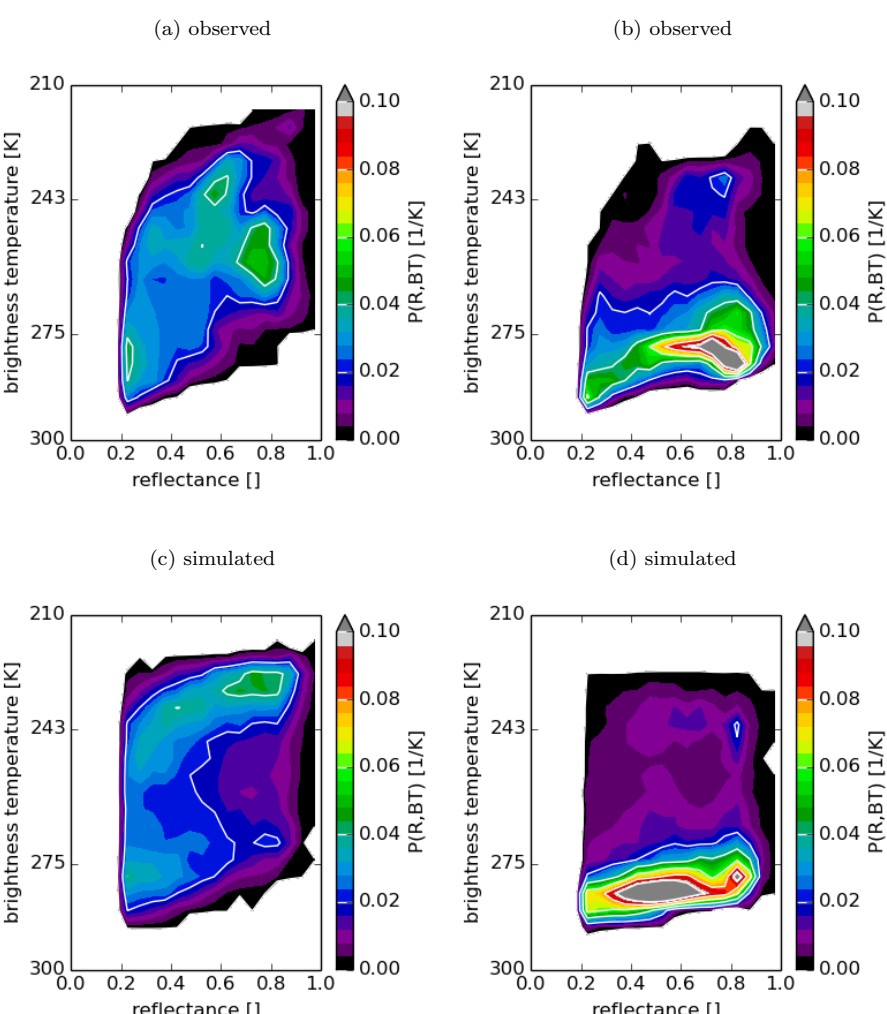

**Figure 6.** Cloud PDF of observations (top) and simulations (bottom), normalized by the sum as a function of brightness temperature and solar reflectance at 29 May (left) and 02 June 2016 (right) at 12 UTC.





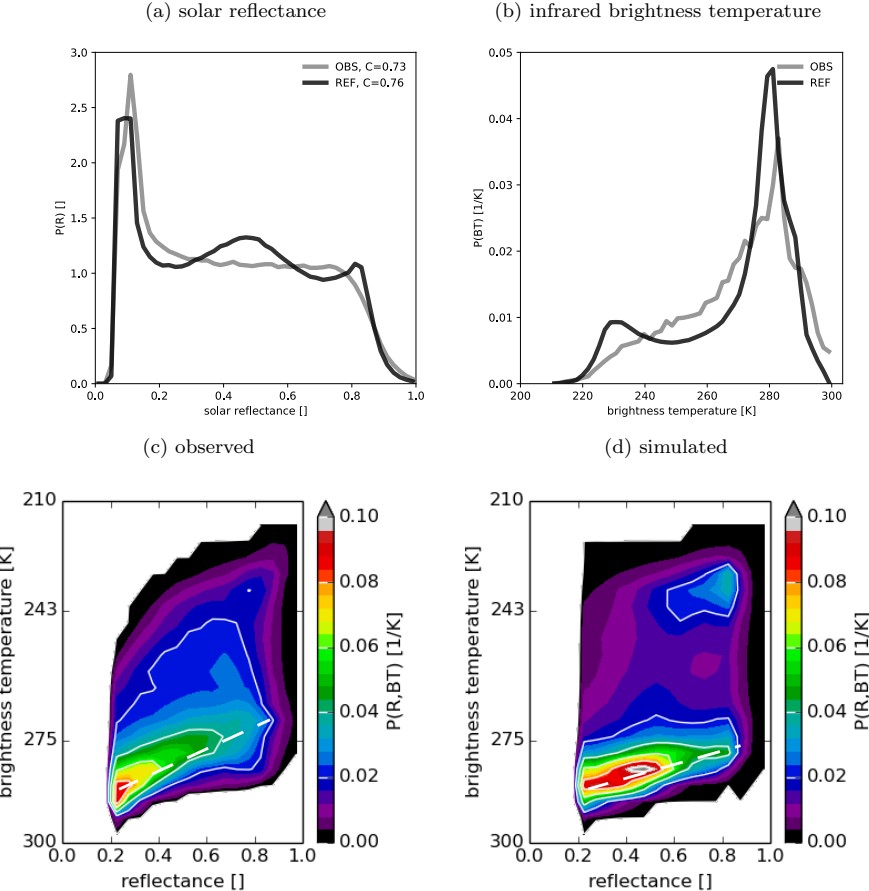

**Figure 7.** Climatology of VIS006 solar reflectance (a) (with cloudiness in the legend), IR108 infrared brightness temperature (b) and cloud PDF (bottom) of observations (c) and simulations (d) at 12 UTC.

clouds on the 29 May. The combined histograms (Fig. 6a & 6c) provide the additional information that this overestimation of clouds mostly happens for thick clouds (R>0.6). This indicates that the model produced too strong deep convection. On 02 June, where lower clouds dominated the scene, the observation and simulations agree on the vertical location of the shallow
cumulus clouds (Fig. 6b & 6d). However, solar reflectances are primarily distributed around 0.7 in the observation and around 0.5 in the simulation. Compared to the 1D reflectance histogram, the 2D PDF provides the additional information that the systematic reflectance errors are related to low clouds. These two days with predominantly deep convective clouds (29 May) and low clouds (02 June) are exemplarily for different cloud types and formation processes. Their analysis therefore illustrates the benefit of combining a solar and an infrared channel.



## 3.2 VIS006 and IR108 climatology

The analysis of individual cases presented above illustrates certain characteristics, but longer periods are required to identify systematic model deficiencies. To address this, we now present results for the 30-day period. The observed mean VIS006 solar reflectance distribution at 12 UTC reveals a clear-sky peak at low reflectance values ($R \in [0, 0.2]$), a nearly uniform distribution for higher reflectances ($R \in [0.2, 0.8]$) and a sharp decrease for reflectances higher than 0.8 (Fig. 7a). The distribution of the reference simulation overall looks similar, but shows some deviations from the flat plateau seen for the observations, with a surplus of clouds around a reflectance 0.5. Fig. 7b presents a histogram of the 30-day mean IR108-BT at 12 UTC and overall confirms findings from previous studies using convection-permitting models. There are generally too many clouds with low brightness temperatures (BT<240K). This, together with an underestimation of mid-level clouds in our ICON simulations is a well known issue that has also been found in other studies (e.g. Illingworth et al., 2007; Pfeifer et al., 2010; Böhme et al., 2011; Keller et al., 2016). The distribution further reveals a clear-sky bias, where the model underpredicts high BT values.

In general, the cloud climatology based on 2D PDF plots indicates that the model and observation distributions have similar structures (Fig. 7c & 7d). Noticeable differences in the distribution occur in boundary-layer clouds. The increase in solar reflectance with decreasing brightness temperature (increasing height) is noticeably steeper in the observations (indicated by dashed white lines in the plots). This means that thick boundary-layer clouds consistently reach higher levels in the observations, and suggests that shallow convection is too weak in the model. The 2D PDFs further indicate that the surplus of clouds around a reflectance of 0.5 in the model is related to boundary layer clouds, revealing a deficiency in the model representation of liquid water clouds. In addition, the simulation lacks in producing enough mid-level clouds at all solar reflectances. Finally, a secondary maximum at low BTs and high solar reflectance ($R \approx 0.8$) is apparent in the simulations but not in the observations. This maximum mainly corresponds to deep convective and precipitating clouds, which are either too active or produce too much ice, similar to 29 May. High-level clouds (cirrus as well as iced cloud tops) and low-level clouds are generally overestimated.

The combined histograms clearly show important shortcomings in shallow and deep convection. Combined histograms can thus provide additional information on the nature of the systematic errors evident in the 1d histograms, and very valuable information for model development, showing which model configuration produces more realistic clouds.

## 3.3 Global horizontal irradiance

In the previous section, we examined systematic deviations between observed and simulated VIS006 solar reflectance and IR108 brightness temperature. Now we investigate if improvements in the forecast of satellite solar reflectance are related to improvements of irradiance forecasts at the surface, which are crucial for the prediction of PV power production. For an effective cloud albedo (CAL, Mueller et al., 2011), the basic relation between surface solar irradiance (SSI) and CAL is predominantly linear, assuming energy conservation (Cano et al., 1986; Beyer et al., 1996). CAL is based on a broadband visible channel and SSI is the downwelling broadband solar irradiance. However, using one narrowband visible channel the dependency can not be exactly linear because of several reasons: A high amount of incoming solar energy is absorbed in the atmosphere by water vapour and ozone, which is not represented in VIS006 solar reflectance, since its wavelength is centered





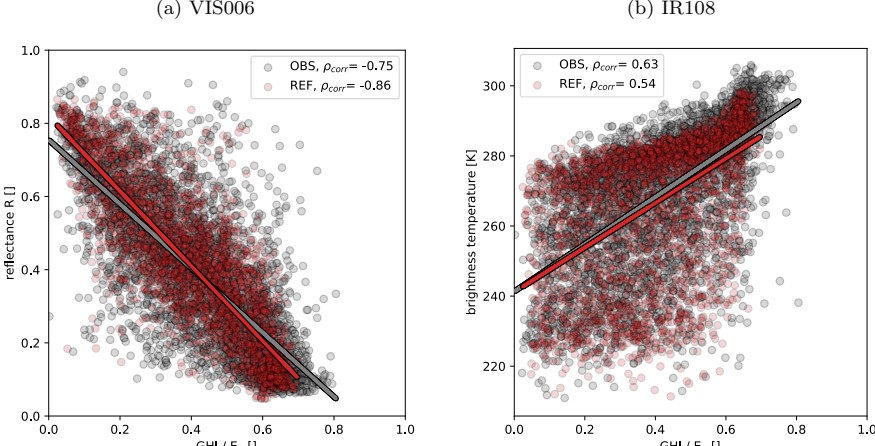

**Figure 8.** VIS006 solar reflectance (a) and IR108 brightness temperature (b) against fraction of incoming global horizontal irradiance (GHI/$E_0$) at 12 UTC. Here, $E_0$ is assumed to be 1367 W/m$^2$ and the number of matches is 3365.

in the atmospheric window. Additionally, the solar reflectance also depends on sun-satellite geometry and 3D RT effects. Moreover, 3D RT effects also influence the irradiance at the surface (e.g. by cloud enhancement). Still, one would expect

reflectance to be anticorrelated with GHI. The incoming portion of GHI (GHI divided by solar constant $E_0$= 1367 W/m$^2$) and VIS006 solar reflectance at pyranometer stations indeed reveals a strong negative correlation, with $\rho_{obs}$ = -0.75 and $\rho_{sim}$ = -0.86, respectively (Fig. 8a). Theoretically, we can expect a higher correlation between these two quantities. However, the observed values differ in timing as GHI is an integrated quantity over the last hour, whereas VIS006 solar reflectance is an instantaneous value. This reduces the correlation due to advection, formation or dissipation of clouds during the observation

time window of GHI. Additionally, aerosols are neglected in our reference simulation and with standard operator settings. A rough estimate of the contribution of aerosols on solar reflectance distribution is provided in Sect. 4.2.

For the infrared channel, one would expect BT to be primarily anticorrelated with cloud top height, and not the optical thickness of the clouds, which determines how much radiation reaches the surface. However, as many high clouds are caused by convection, there should still be some correlation between BT and GHI. Figure 8b shows the dependence of GHI on the

IR108 brightness temperature with correlation coefficients of 0.63 and 0.54 for observations and simulation, respectively. As expected, correlation values are lower than for VIS006, and additionally, it should be noted that the correlation of brightness temperature and GHI may be reduced for other weather situations and less convective periods.

These results indicate that reducing the error of synthetic satellite images, in particular for visible satellite channels, should improve radiation forecasts.





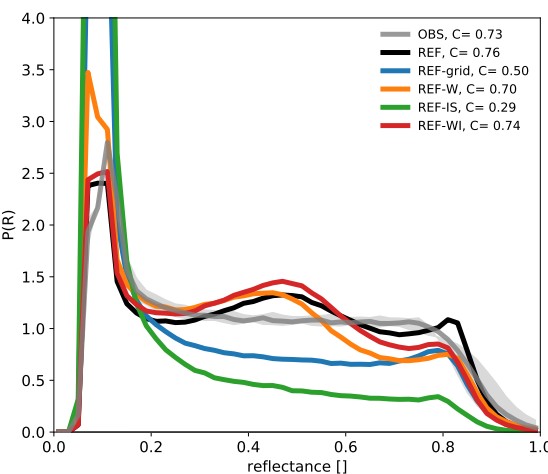

**Figure 9.** Visible reflectance histograms for the test period computed for the observations (OBS) and the reference experiment (REF). The additional distributions were computed using only the grid-scale clouds (REF-grid), onyl the water clouds (REF-W) and only the ice clouds (REF-IS) of the reference experiment, respectively. For the red line (REF-WI) water and ice clouds are taken into account and only the snow contribution to the ice clouds was omitted. The numbers in the legend indicate the cloudiness, i.e. the fraction of pixels exceeding a relfectance of 0.2.

## 4 Sensitivity of synthetic VIS006, IR108 satellite and surface irradiance observation

### 4.1 Contributions of different clouds to the reflectance distribution

For understanding the sensitivity of the synthetic visible satellite images to changes in operator settings and model modifications, it is helpful to determine the contribution of different hydrometeor types and subgrid-scale clouds to the reflectance histogram of the reference run (Fig. 7a). Figure 9 shows the observed and simulated VIS006 solar reflectance distribution (OBS and REF are the same as in Fig. 7a), the distribution that results from taking only grid-scale clouds into account (REF-grid) and several distributions obtained by using only certain types or combinations of hydrometeors.

Grid-scale clouds only lead to a distribution with a nearly flat plateau between reflectances 0.3 and 0.7, a feature that is also found in the distribution of the observed reflectances. However, the fraction of cloud pixels would decrease from $C = 0.76$ to 0.5 if only grid-scale clouds were present. Adding subgrid clouds results in much better agreement with the observed value of $C = 0.73$. It is thus essential to take these additional subgrid clouds into account. However, the imperfect parameterisation of subgrid clouds also contributes to deviations in the shape of the distribution: While the distributions of the observations and the grid-scale clouds only simulation exhibit a relatively flat plateau, the addition of subgrid clouds leads to a histogram curve with a pronounced maximum at 0.5 and a minimum at 0.7.

When only water clouds are used as input to the operator (REF-W), the cloudiness falls off from $C = 0.76$ to $C = 0.70$. Primar-





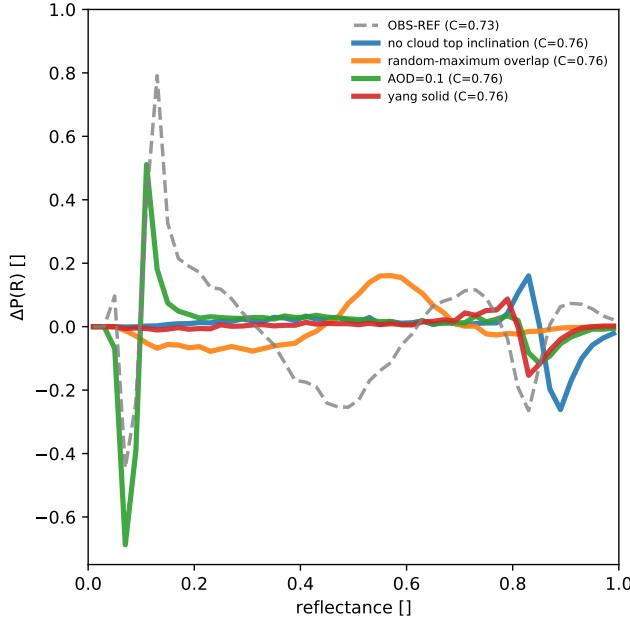

**Figure 10.** Differences between reflectance histograms obtained for the reference run with modified operator settings and standard settings. The modified settings are switching off the cloud top inclination, using maximum-random instead of random subgrid cloud overlap, including aerosols with an optical depth of 0.1 and changing the cloud ice particle habit to solid columns. For comparison also the difference between observation and reference run histogram is shown (dashed curve).

ily, reflectances larger than 0.5 become slightly less frequent. In contrast, taking only ice clouds (including snow) into account (REF-IS) has a more substantial impact on the histogram and results in much smaller cloudiness of $C = 0.29$. Water clouds thus play a much more substantial role for the reflectance distribution than ice clouds. This result is not surprising as the total column ice content is much smaller than the water content (Fig. 2c) and additionally larger ice particles are less effective in scattering light than smaller water droplets (Fig. 2a).

In both the water-only and the ice-only cases, the corresponding subgrid clouds are included. The water-only curve (REF-W) shows the same deviation from the plateau-like shape of the observed distribution as the curve computed for all clouds (REF), but the ice-only curve (REF-IS) does not. Thus, it seems that the subgrid water cloud parameterisation needs to be improved to get better agreement in the histogram shapes. Finally, ignoring the simulated snow content (REF-WI) has a small, but detrimental effect. This emphasizes the need for including snow in the computation of the RT input variables as discussed in section 350 2.3.

## 4.2 Estimated uncertainty of VISOP

Forward operators use fast, approximate RT methods and rely on the limited information that is available from the NWP model. Due to missing RT effects and missing information (e.g. on subgrid-scale cloud properties) their output is to some extent





uncertain. While forward operators for thermal infrared channels have been available for some time and their uncertainties
have been investigated in several studies (e.g. Senf and Deneke, 2017; Saunders et al., 2017, 2018), no such information is
available for visible channels. In the following, the uncertainty related to what we regard as the most critical error sources will
be estimated by varying the corresponding operator settings.

The potential sources of uncertainty to be investigated are related to missing 3D RT effects, unknown or inconsistent overlap
statistics of subgrid-scale clouds, the spatial and temporal variation of aerosols and the shape of cloud ice particles. To estimate
upper limits of the uncertainty in the reflectance distribution related to these sources, we repeated the computation of visible
reflectances applying VISOP to the reference simulation with deactivated cloud top inclination (CTI) parameterisation, random
instead of random-maximum subgrid cloud overlap, and aerosols or a different kind of ice habit included in the MFASIS LUT.
The deviations in the reflectance distribution for the reference run caused by changing these operator settings are shown in
Fig. 10.

The subgrid cloud overlap assumptions would not be a source of operator uncertainty if the assumptions in the NWP model
and the operator were entirely consistent. However, the near-operational version of ICON employed to perform the model
runs for this study uses inconsistent overlap assumptions in the infrared and visible part of the spectrum. This inconsistency
will likely be corrected in future versions, but at the moment, it means that the operator cannot be entirely consistent with the
model. The deviation in the reflectance distribution caused by changing the assumption from maximum-random to random in
the operator (orange line in Fig. 10) can be regarded as an upper limit for the impact. Changing the assumption shifts the peak
around R=0.5 (which is related to subgrid clouds, as discussed in Sect. 4.1) to higher reflectances, but has not much influence
on reflectances larger than 0.7.

Missing or imperfectly modelled 3D RT effects are likely the source of uncertainty that is most difficult to quantify. According to Scheck et al. (2018) the most important 3D effect is related to the inclination of the cloud top surface, which influences
the observed reflectance. The parts of the cloud top surface tilted towards the sun appear brighter and those tilted away from
the sun darker. The cloud top inclination correction (CTI, see Scheck et al., 2018) accounting for this effect has been shown to
reduce the error with respect to full 3D RT calculations and is included in the reference run. The main effect of the CTI on the
reflectance histogram is to reduce the slope at the high reflectance end of the distribution and to bring it in better agreement
with observations. Switching off the CTI leads to a too steep decline of the distribution at high reflectances, which is visible as
a double peak structure at $R > 0.8$ in Fig. 10. Other 3D RT effects like cloud shadows may also play a role, in particular for
larger zenith angles. However, by focusing on observations near local noon, their influence should be minimized.

According to retrievals based on measurements at AERONET stations (see Giles et al., 2019) in Germany, the mean AOD
in June 2016 was in the range 0.06 to 0.12 at a wavelength of 675 nm, which is similar to the wavelength of the visible
channel considered here. To estimate the impact of these aerosols on the reflectance histogram, an MFASIS LUT was computed
that includes aerosols (the "continental clean" aerosol mixture available in libRadtran, see Emde et al. 2016) with an optical
depth of 0.1. Including aerosols in the MFASIS LUT, i.e. taking direct aerosols effect into account, influences the reflectance
histogram in two ways. Aerosols can scatter photons out of their path towards the satellite, which is the dominant effect at
high reflectances, or scatter photons towards the satellite that would otherwise not have reached it, which is important for low





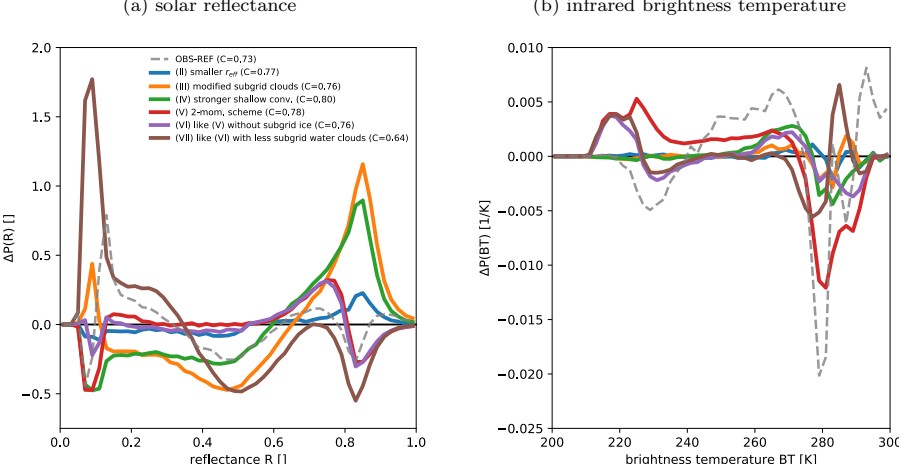

**Figure 11.** Deviation of VIS006 solar reflectance (left) and IR108 infrared brightness temperature (right) PDFs of perturbed model simulations from the ones for the reference run. For comparison also the difference between observation and reference run is shown (dashed curve).

reflectances. In the presence of aerosols the high reflectance end of the distribution is thus shifted towards lower reflectances
and the low reflectance end towards higher reflectances. Shifting the pronounced ground peak in the distribution causes a double peak structure at low reflectances in Fig. 10, whereas shifting the flat high reflectance end only causes a single negative peak. In general, the error introduced by direct aerosol effects for events like (Saharan) dust outbreaks can be higher, and could potentially lead to significant errors in solar reflectances. Days affected by such events, which did not occur during our test period, should thus be excluded from model evaluation studies.

Finally, the shape of cloud ice particles is also an uncertain factor that influences the reflectances distribution. Changing the shapes quite strongly from the `baum_v36` general habit mixture (Baum et al., 2014) to solid columns (using the optical properties by Yang et al. 2005) basically only affects the highest reflectances, which are slightly reduced. The ice habit is thus not likely to cause large uncertainties in the reflectance distribution.

## 4.3 Sensitivity to model settings

Figure 11 shows the deviations of the reflectance and BT distributions computed for model runs using modified settings (see Sect. 2.1) with respect to the reference run. In general, these deviations are of similar magnitude as the systematic deviations between the observations and the model equivalents for the reference run discussed in section 3 (see dashed curve in Fig. 11). In section 3, we identified several reasons for systematic deviations between the simulation and observations: An underestimation of thick clouds (R in [0.6,0.8]), a too low boundary layer height, too many high clouds and an insufficient representation of
low-level water clouds. As further analysed in Sect. 4.1, we found that the discrepancy in low-level clouds mainly arises from subgrid water clouds (R in [0.3,0.6]).





Figure 11a shows the effect of model modifications on the reflectance distribution. The first modification (experiment II), reducing the effective radii by increasing the updraft velocity and thus also the number of cloud condensation nuclei, leads to more thick clouds with $R > 0.7$ and less thin clouds with $R < 0.5$. Changing the subgrid cloud parameters (experiment
III) or reinforcing shallow convection (experiment IV) has a qualitatively similar but much stronger impact on the reflectance distribution. Pixels with dense clouds become more numerous and the number of pixels with thin to medium clouds is reduced. These changes are larger than the deviations of the reference run (experiment I) from the observation (dashed line in Fig. 11a). In case of the modified shallow convection, the cloudiness increases from 0.76 to 0.8, which means that the deviation from the observed value of 0.73 is considerably larger.

Switching to the double-moment microphysics scheme (experiment V) mainly moves pixels with very high reflectances ($R > 0.8$) to somewhat lower reflectance values between 0.6 and 0.8 and increases the cloudiness slightly. Thin to intermediate clouds ($0.2 < R < 0.6$) are only weakly affected. Still using the two-moment scheme but turning off subgrid-scale ice clouds (experiment VI) slightly decreases the cloudiness but basically leads to the same distribution as experiment V. Hence, ice subgrid-scale clouds cannot be responsible for the surplus of pixels with solar reflectances around $R = 0.5$ that was attributed
to subgrid clouds in Sect. 4.1. Finally, reducing the subgrid-scale water clouds (experiment VI) in addition leads to much larger changes, with negative peaks around $R = 0.5$ and $R = 0.8$ and positive values for $R < 0.35$. These changes point into the right direction to mitigate the deviations of the reference run (dashed line in Fig. 11a). However, here the modification is too strong as cloudiness is dramatically underpredicted in this case ($C = 0.64$). Compared to visible reflectances, the changes in the BT distribution introduced by modified model settings are more difficult to interpret as the cloud top height is an
important additional parameter. The modifications in experiments II and III only affect water clouds and thus only lead to changes at higher BTs. These changes are relatively small compared to those required to correct the deviations of the reference run (dashed line). Making shallow convection stronger (experiment IV) has a stronger impact and increases the number of pixels with BT between 250K and 275K at the expense of those with higher values. Switching to the double-moment scheme (experiment V) increases the number of middle to very high clouds for BT<270 K, and introduces a substantial reduction of
the clear-sky and low-level cloud signal (BT around 280 K). These changes indicate that the two-moment scheme generates even more dense ice clouds than the one-moment scheme in the reference run, which already predicts too many of these clouds. These high clouds obscure lower clouds and the surface, which leads to less pixels with high BTs. Switching off subgrid ice clouds in the two-moment simulation (experiment VI) reveals that the peak around BT = 220 K is related to grid-scale clouds in the double-moment scheme, and the distribution of middle clouds is more like the single-moment simulation. Additionally
modifying the subgrid liquid water clouds (experiment VII) again mainly affects the clear-sky and lower-level cloud signal.

Comparing the changes in the reflectance and BT distribution that were introduced by modified model settings within their estimated uncertainty leads to the following interpretation: The reflectance distribution is mainly affected by changes to water clouds and is only weakly influenced by changes to ice clouds. In contrast, the BT distribution is most strongly affected by changes in the ice clouds, but modified water clouds also have some influence on higher BTs. The distinct changes in the
distributions caused by the individual model modifications allow to assess which modification could be useful to mitigate deviations from the observed distributions.





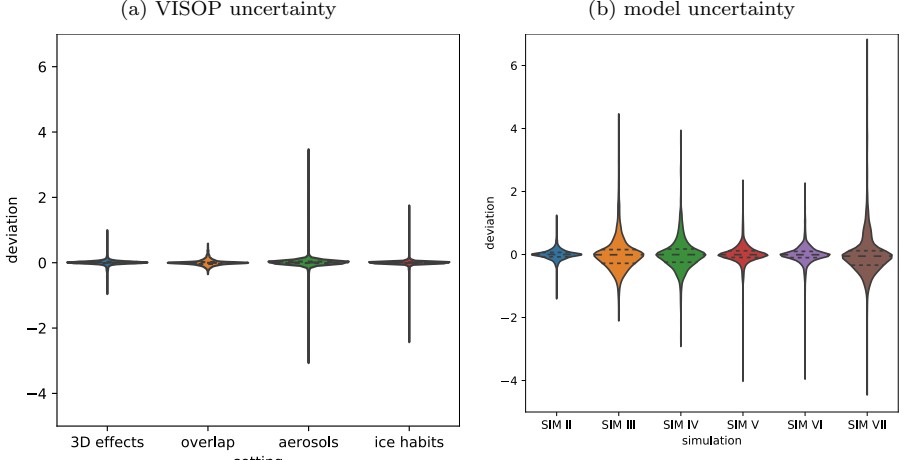

**Figure 12.** Distributions of **bin-by-bin** differences from reference run. Horizontal dashed lines indicates $25^{th}$, $50^{th}$ (median), and $75^{th}$ percentile.

The results shown in Fig. 11a indicate that a modified version of experiment VII with weaker modifications or a combination of II, III and IV could be able to achieve the corrections required for the reference run, i.e. to reproduce the dashed line (OBS-REF). In both cases the subgrid water clouds play an important role. To correct systematic errors in the reflectance distributions

it therefore seems particularly important to tune or advance the subgrid cloud scheme. While the reflectance distribution is not sensitive to changes in subgrid ice clouds, these are clearly important for the BT distribution (compare experiments V and VI in Fig. 11a,b). The combined information from the two parts of the spectrum can thus provide guidance on optimizing the subgrid cloud scheme.

In contrast to visible reflectances, there is no obvious way to scale or combine the model modifications in order to eliminate

the errors of the reference run, i.e. to reproduce the dashed line in Fig. 11b. Additional or different model modifications appear to be required for this purpose, but the results presented here already indicate that particular modifications leading to less grid-scale ice clouds are required.

## 4.4 Sensitivity intercomparison for visible reflectances

The comparison of Fig. 10 and Fig. 11a already indicates a considerably larger effect of model modifications compared to that

of operator uncertainties on the reflectance distribution for the full test period. To provide a clearer comparison of the impact of model modifications and operator uncertainties, we computed the individual changes on each day of the test period in all of the reflectance bins (see Sect.2.5). The violin plots in Fig. 12 show these daily bin-by-bin deviations of the reflectance distribution caused by changes in the operator settings and model modifications. Figure 12 indicates that also for individual days of the test period the changes due to model modifications are much larger than the ones related to operator uncertainty. The median

deviation and the interquartile range (difference of 75th and 25th percentile) are about one order of magnitude larger for the





model uncertainty. As already mentioned, aerosols will have a much stronger impact during e.g. dust events, but such events should not be included in test periods for model evaluation.

In general, the operator uncertainties are thus a second-order effect compared to model modifications. Visible satellite images are therefore well-suited to detect and overcome model deficiencies and to provide guidance for model tuning. Still, some of
the deviations of the model reflectance distribution could be related to deficiencies of the operator. An improved cloud top inclination or changes in the cloud ice optical properties could mitigate some of the deviations at high reflectances and using the correct aerosol optical depth can particularly improve the low-reflectance end of the distribution (see Fig. 10). However, for a broad range of reflectances between 0.2 and 0.8 it is only the inconsistency in the overlap assumption that makes the operator results uncertain. As discussed above, this is actually only a temporary issue related to the current versions of the
ICON model. As soon as the overlap assumptions in the the model are consistent, the correct choice of the overlap assumption can be regarded as a model setting and model evaluation using visible reflectances can provide information on suitable choices.

## 5   Conclusions

We investigated systematic differences between satellite observations and corresponding synthetic observations from the pre-operational ICON-DE model to understand better the representation of clouds and radiation in NWP models. For this purpose,
a semi-free 30-day convection-permitting hindcast simulation was conducted that is only forced by low-resolution analysis boundary conditions for a highly convective period in May/June 2016. Besides, some additional simulations with modified model settings were conducted to identify dominant error sources and identify potential approaches for improving the representation of clouds in ICON-D2.

The evaluation facilitates a novel approach based on both solar and infrared satellite observations. The combination of obser-
vations in these two spectral ranges provides significantly more and complementary information than the use of only infrared observations pursued in previous studies. While infrared observations provide information on cloud top height, their signal quickly saturates in the presence of clouds. This means that infrared observations can only distinguish a small range of cloud water contents and information on water clouds may be obscured by cirrus clouds above. In contrast, solar channels are relatively insensitive to ice clouds and can distinguish a much more extensive water content range.
As solar satellite observations are novel for model evaluation, we conducted a number of sensitivity experiments with modified operator settings to investigate the recently developed forward operator's uncertainty transforming from model to observations space. The comparison revealed that the operator uncertainty is roughly one order of magnitude smaller than the sensitivity of the results to modified model settings. This further emphasises the usefulness of solar channels for model evaluation and improvement.
In addition, we investigated the correlation of these two satellite channels with irradiance in view of improving forecasts of photovoltaic (PV) power production. This revealed a strong negative correlation of solar satellite observations with surface irradiance, which means that such observations are well-suited to improve model cloud parameterisations for better PV power production forecasts. In contrast, infrared observation only shows a moderate correlation to surface irradiance and irradiance



observations themselves would be too sparse to provide suitable statistics for a detailed model evaluation.

The combined use of solar and infrared observations allowed to identify specific model deficiencies, e.g. too many high cirrus clouds, too weak shallow convection, deficiencies in the model representation of subgrid clouds, too strong deep convection or too much-related production of cloud ice. Several model sensitivity experiments targeted these deficiencies and point towards potential approaches for model improvement. However, solving these challenging issues will require additional studies given the number of interacting processes that contribute to the formation, modification and dissipation of clouds. Nevertheless, it is

of utmost importance to advance the representation of clouds and radiation for the use of cloud-affected satellite observations in data assimilation, the prediction of PV power production, and last, but not least accurate climate simulations.

*Data availability.* The relevant research data (observations and model equivalents) is publicly available at https://doi.org/10.5281/zenodo.4548922 (Geiss et al., 2021). The model simulation output will be archived at LRZ for ten years.

*Author contributions.* All authors have contributed equally

*Competing interests.* The authors declare no conflicts of interest.

*Acknowledgements.* Funding for this research is provided by the integrated project MetPVNet, project number 0350009A, financed by the Federal Ministry for Economic Affairs and Energy. The authors would like to thank the Hans Ertel Centre for Weather Research (Weissmann et al., 2014; Simmer et al., 2016) for supporting this work. This German research network of universities, research institutes and the German

Weather Service is funded by the BMVI (Federal Ministry of Transport and Digital Infrastructure).



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
