# Peer review of "Understanding the model representation of clouds based on visible and infrared satellite observations"

_Atmospheric Chemistry and Physics, 2021_

## Author Comment (AC1)

**Response to Referee Comment (RC1) on**

Understanding the model representation of clouds based on visible and infrared satellite observations (https://doi.org/10.5194/acp-2021-5)

We appreciate the detailed comments of the reviewer and have revised the manuscript to address all remarks. Our response is provided in black and the review in blue.

*The paper discusses biases in the representation of clouds in convection-permitting simulations with the ICON-D2 model. The authors use a combination of visible satellite reflectances and infrared brightness temperatures to derive model shortcomings. Using satellite forward operators, observation equivalents are computed from model data which allow for a direct comparison with observations. The authors contrast uncertainties in the visible forward operator to sensitivities in model parameter setting. Based on their analysis result, the authors emphasize that the assumptions on subgrid-scale water clouds are the primary source for model biases in the visible spectrum and that the representation of these clouds need to be carefully revised to make further improvement possible.*

*I think the present study will become a valuable resource and I recommend the publication of the manuscript in ACP after major revision.*

**General Remarks**

*The paper is in general well written and structured. The objectives are clearly stated and all arguments are well supported. I don't see that language usage is of any concern. The manuscript discusses a relevant topic in atmospheric research, advanced analysis techniques are applied and the resulting scientific outcome is of interest for a wider audience.*

**General Comments**

*\* Relationship to ISCCP-style analysis?: Combining VIS and IR as joint histograms or PDFs is not a new technique. There are a lot of different examples in literature which use joint histograms of cloud-optical thickness and cloud-top height to assess quality of climate model, global and regional weather forecasts. Two respective examples are: Zhang, M. H., et al. (2005), Comparing clouds and their seasonal variations in 10 atmospheric general circulation models with satellite measurements, J. Geophys. Res., 110, D15S02, doi:10.1029/2004JD005021. & Otkin, J. A., & Greenwald, T. J. (2008). Comparison of WRF model-simulated and MODIS-derived cloud data. Monthly Weather Review, 136(6), 1957-1970. and much more references therein and also based on these papers. It feels like you had completely overlooked this branch of studies and their*

*relationship to your research. Please add a comprehensive discussion on this topic in your introduction and in your results section (where it is appropriate).*

Thank you for pointing to these additional references, which we clearly should have cited. We have added a discussion of these approaches to the introduction and cite also some of the studies in the results section.

While these studies do indeed use information derived from infrared and visible channels, all of them seem to be based on retrieved quantities. The novel aspect of our study is to perform the analysis directly in observation space. We performed another literature search, but could not find previous studies using this observation-space approach for visible and IR observations.

We believe that using forward operators is a favourable approach over the use of retrievals as characterizing the retrieval errors can be problematic (Errico, R. M., Bauer, P., & Mahfouf, J., 2007, https://journals.ametsoc.org/view/journals/atsc/64/11/2006jas2044.1.xml ). Obviously a good characterization of errors is required also for the evaluation of changes in the model and we see this as a clear advantage of our approach.

Actually, our approach is fully in line with the recommendations from the 4th ECMWF workshop on assimilating satellite cloud and precipitation observations for NWP ( https://www.ecmwf.int/sites/default/files/medialibrary/2020-06/Working_Group_Summaries_2020.pdf ), where it is argued that it is easier to quantify errors for forward operators than for retrievals and that it is recommended to perform a more comprehensive and systematic evaluation of the errors in forward operators.

*\* Connection to solar power prediction: To my opinion, the analysis that tries to establish a connection between satellite data and global irradiance measured at surface is the weakest part of your manuscript. I guess you try to make the argument that solar power prediction would improve if the representation of clouds (measure from space) is becoming more realistic. However, your analysis and the presented arguments do not support such a conclusion by now. I recommend you to revise the analysis in Sect. 3.3. It would be beneficial for the reader to show how biases in GHI are correlated with the biases in VIS and IR108. One would expect that lower GHI biases coincide with lower VIS biases which would support the conclusion that the use of visible satellite data is beneficial for ground-based irradiance predictions.*

We see the point that section 3.3. was not well-connected to the rest of the paper and did not provide a detailed investigation of this connection. Thus, we removed 3.3. instead added version of the GHI-VIS / GHI-IR correlation plot containing only observations as a motivation in Section 2.

*\* Figure Quality: Please make sure that font sizes in your figures (e.g. axis labels, legends) are sufficiently large.  Text in figures should not be significantly smaller than the text in the figure caption. Please update your figures accordingly!*

All plots were updated.

**Detailed Comments**

*L. 11: "modified ...settings": Please rephrase to make more clear that both, variations in model settings and forward operator uncertainties, have been considered.*

We have rephrased the abstract and introduction to address this.

*L. 16: "VIS solar reflectance and global horizontal irradiance": Please make clear that the former in measured at TOA and the latter at surface.*

We removed the statement.

*L. 17: "will coincide" -> "can enable"?*

We removed the statement.

*L. 35: "are usually ... smaller" - Please support this statement with references!*

We removed the statement.

*L. 45: "minimization" -> reduction*

Changed accordingly: minimization → reduction

*L. 46/47: "Unfortunately, ...." Statement is very general and for sure not true for all current NWP systems. Please make it more specific and supported by references!*

We added two references on this.

*L. 51: "meteorological sensitive areas": Unclear what this means.*

We added atmospheric instability for clarification.

*Fig. 1: Does not appear to be referenced in the right order. Labels are too small.*

We added a reference in section 2.1; all Figures were updated.

*L. 58: "solar irradiance fluctuation" + "at surface" (or at ground). Also this statement needs to be support by a reference.*

We added a reference in the rephrased introduction.

*L. 65 full paragraph: This needs like a conclusion paragraph and is not in the right place here. Please rephrase and complete paragraph. This is the place where you can state your research question and outline how approach your research goal.*

We have rephrased the introduction, including this paragraph.

*L. 78: "cloud climatology": Here, and everywhere else: Please avoid the term cloud climatology which is mis-leading because it refers to long-term (!) cloud statistics which is not the case in your study. Please use "time-mean statistics" (or "time-average") instead.*

We do agree and have revised accordingly in the whole paper. Instead of climatology we use "cloud statistics for a full test period" and "statistics for the full period".

*L. 86: "ICON-D2". Please name the model version here.*

We added the information (a development version based on version 2.6.1).

*L. 96: "We performed six" + "additional"*
Revised accordingly.

*L. 98: "The objective was to ..." Please rephrase sentence.*

*We rephrased the sentence.*

*eq. (1): needs more explanation! I guess this is only a partial contribution to total cloud cover (might be indicated by subscript cc_turb). Is this cc contribution just added to the other contributions? What is q_sat? And where does the scheme come from (reference) and how should it be interpreted? Parameter B needs to be explained as well.*

A more extensive description of the cloud-cover scheme has been added to the paper in Sect. 2.1, exchange point 2

*L. 125: "like the operational one." -> "like the operational one-moment scheme."*

Revised accordingly: one → one-moment scheme

*L. 129: "cloud-concentration number" -> cloud droplet number concentration"*

Revised accordingly: cloud-concentration number → cloud droplet number concentration

*L: 135: 2*10^4 to 4*10^4 hPa: This is definitely too large! Wrong units?*

Thanks for spotting this error, the unit was indeed wrong (Pa instead of hPa).

*L. 146: "visible 0.6 um channel" Please specify if the visible reflectance is corrected by solar zenith angle. If yes, comparison in Fig. 8 would be inappropriate because GHI is scaled by a constant.*

Thanks for this point. Now, GHI is corrected by solar zenith angle (added it in the caption and x-label of the Figure) and we only show the observed dependency as a motivation.

Figure 8 → Figure 3

*L. 151: "TCW" / "TCI": I would prefer "LWP" and "IWP", liquid-water path & ice-water path is more commonly used.*

Revised accordingly to LWP and IWP

*Section 2.3 misses to tell how aerosol is treated.*

We added a sentence for clarification

*Eq. (2):*

*\* Consistency of symbols: You use low-case q in eq. (1) for content. And you use capital R as reflectance later. I suggest to use consistent symbols.*

Revised accordingly, also in the text;

*\* Is this equation consistently applied to visible and infrared? Please comment on this aspect.*

We applied this "effective effective radius" only in the visible forward operator, as it was not possible to use an external effective radius for water with RTTOV 12.1, which was used for the infrared calculations. Interestingly, we have recently performed tests with the new version of RTTOV (13.0), which allows for using the model effective radius. The differences in the histograms arising from using different effective radii were significantly smaller in the infrared channels than in the visible ones.

*\* How does this method compares to the generalized effective diameter in Senf and Deneke (2017), AR, eq. (B.3)?*

As in Senf and Deneke, the generalized effective radius is based on the addition of the volume extinction coefficient from the different phases. Our more simple formula is the result from assuming the same shape for ice and snow.These details are now included in the text.

Changes to the text:

new Line 207: […] and snow $r_{s,eff}$. The effective radii for ice and snow are calculated under the assumption that both hydrometeors behave as randomly-oriented needles, and using the mass-size relationships, size distributions and number concentrations from the microphysics (for details see Fu et al. 1997 and Muskatel et al. 2021).

new Line 211: [...] snow phases, similar as Senf and Deneke (2017).

References:

Fu, Q.; Liou, K.N.; Cribb, M.C.; Charlock, T.P.; Grossman, A. Multiple Scattering Parameterization in Thermal Infrared Radiative Transfer. *J. Atmos. Sci.* **1997**, *54*, 2799–2812

Muskatel, H.B.; Blahak, U.; Khain, P.; Levi, Y.; Fu, Q. Parametrizations of Liquid and Ice Clouds' Optical Properties in Operational Numerical Weather Prediction Models. *Atmosphere* **2021**,*12*, 89. https://doi.org/10.3390/ atmos12010089

Senf, F., & Deneke, H. (2017). Uncertainties in synthetic Meteosat SEVIRI infrared brightness temperatures in the presence of cirrus clouds and implications for evaluation of cloud microphysics. Atmospheric Research, 183, 113-129.

*L. 192/193: This is much too short! SGS clouds play an important role in your analysis. Please be much more explicit about your treatment of SGS clouds. What are the assumptions about microphysics (effective radius, adiabaticity) of SGS clouds? How does this impact cloud-optical thickness?*

Both the microphysics and cloud-cover schemes produce a mass concentration qc/qi. The diagnostic qc/qi combines the mass concentration from both schemes. We calculate the effective radii using these mass concentrations and the assumptions for number concentration, probability distribution function and particle mass-size relation form the microphysics scheme.  Therefore there are no differences in the assumptions for the calculation of the effective radius for grid clouds, subgrid clouds or the combination of them. Since we have a consistent calculation for grid and subgrid clouds we have not made any sensitivity study about the optical properties of subgrid clouds.

Changes to the text:

Line 214: […] water or ice. We assume no differences in the microphysical and optical properties of grid and subgrid clouds, so that the effective radius calculation is the same for both cases.

*L. 197: "calibration offset" To my opinion, you are removing a systematic bias from the simulation which is fine in general. However, I would phrase it in that way.*

You are right. We changed it in the text, accordingly.

*L. 201: "spatial resolution" -> Please move to Sect. 2.2.*

The sentence on spatial resolution was moved to Sect. 2.2

*Sect. 2.4: What is the accuracy of GHI measurements?*

This depends on the sensor:

1. CM11 (21 stations), CM21 (5 stations): WMO secondary standard instrument, where error should be less than 2 %
2. SCAPP (96 stations) deviates less than 10 % from secondary standard instrument

We think for motivating the work, the sensors are accurate enough.

*L. 215: "... without coarsening and thinning" -> unclear*

We removed this sentence.

*L. 220: I don't understand why you don't take the observation as a reference: eps = P(SIM) - P(OBS)?*

We want to study the effect in simulated reflectances. Observations are only of secondary interest here.

*Violin plots: I would recommend to skip the distribution outside a certain range (<10th and >90th percentiles) to increase readability of the plots in Fig. 12. Otherwise these plots are dominated by the extremes.*

Good suggestion, the y-axis is now limited to (-2,2).

*L. 224: CFAD -> reference*

Revised accordingly: Added a reference for CFADs and we also mention the ISCCP-approach here.

*L. 224ff "Standard atmosphere .... ": Don't understand why you choose this distinction. Much more natural would be <273 K, (273 K... 243 K), <243K which would separate liquid, mixed-phase and ice clouds.*

The reason is that we wanted to be close to WMOs cloud type classification: with an altitude of low clouds < 2000 m, middle clouds < 6000 m and high clouds > 6000m.

*Fig. 4 + 5: Please use same projection as in Fig. 1 or the other way around. Please avoid histograms and use PDFs instead as you introduced PDFs as verifcation metrics.*

We use the same projection for all Figures now.

We only show PDFs now.

*L. 248: Fig. 4a & 4b -> wrong reference, 4b shows BTs.*

Revised accordingly b → c
(Fig. 4 → Fig. 5)

*Fig. 6 caption: White lines: What do they mean? "normalized by the sum" -> confusing. If you show PDFs then normalization is not a matter of choice: \int P(BT, R) dBT dR = 1!*

You are right! We show the PDF and the caption was wrong. We removed the white lines from the plot.

*Fig. 7: Observed BTs are higher than 300 K. Is the range > 300 K considered in the normalization of the PDFs?*

We have extended the range to 310 K.
(includes all observed/simulated BTs in [200, 310] K)

*Sect 3.2. Again, avoid the term "climatology".*

Revised accordingly

*L. 282 / L.284: There is a duplicate statement: "findings from previous studies"; "found in other studies" Please rephrase the two sentences.*

We have rephrased the sentences and added some additional references, where the obs to model approach was used.

*Sect 3.3:*

*\* Please see my general comment. What is the general idea of this analysis? I guess you like to show that GHI forecasts can improve when VIS / BT forecast are more realistic, right? Why don't you show the bias in GHI vs. the bias in VIS? Otherwise, the reader get the feeling that plotting hourly average GHI values against instantaneous VIS observations is rather inappropriate (see L. 312 - 14).*

See above: Sect. 3.3 is now in Sect. 2.2

*\* Caption Fig. 8 "number of matches"-> unclear.*

We replaced "matches" by "collocated observations" in the caption.
[number of collocated surface GHI observations @Pyranometer stations and reflectance (TOA)]

*\* Meaning and usefulness of lines in Fig. 8 is also unclear.*

We removed the lines.

*\* Is scaling of GHI consistent with scaling of VIS radiances? See above.*

We made it consistent (see above)

*L. 335: "imperfect parameterization" Again, a clearer description of the microphysics of SGS clouds would help.*

A more detailed description of the parameterization is provided in the new version of the paper (see comment above). Given its simplicity, we think it is an obvious imperfect parameterization.

*L. 337: flat plateau for grid-scale clouds: Would this mean that this VIS bias can be resolved by proceeding to even higher resolutions, e.g. hecto-scale simulations? Could you comment on this? Are there any indications in the literature?*

According to Wood & Field (2011), Fig. 6, which is based on high-resolution satellite observations, 85% of global cloud cover comes from clouds larger than 10km and the cloud cover contribution from clouds smaller than a few 100m is very small:

[Figure]

[ from https://journals.ametsoc.org/view/journals/clim/24/18/2011jcli4056.1.xml ]

Therefore, subgrid cloud parameterization should become unnecessary for hecto-scale simulations. In fact, Heinze et al. (2017) neglected subgrid clouds in their ICON experiments with 150m resolution and found an improved representation of clouds, compared to km-scale models.

However, cloud representation depends on nearly every parameterization in the model (Zhang, M. H., et al. (2005): ,Otkin, J. A., & Greenwald, T. J. (2008), references you suggested and Webb et al., (2001)). We think that subgrid-scale cloud representation is currently the main challenge for VIS bias.

*L. 347: "it seems that the subgrid water cloud parameterisation needs to be improved" -> or its coupling to the VISOP?*

The visible forward operator sees the same clouds as the model, including the subgrid clouds. This is clearly a failure of ICON representing these subgrid-scale clouds correctly, in particular of subgrid-scale clouds in the boundary layer (see Fig. 8).

*L. 353: "missing RT effects" -> unclear*

We have clarified this statement: *"missing RT effects"* -> "missing 3D RT effects"

*L. 382ff: In this paragraph, it is not clear how you treat aerosols in the reference run.*

We have added a sentence in Sect. 2.3

*L. 387: "Aerosol can scatter photons ..." Sentence reads weird. Please rephrase.*

We have rephrased the sentence.

*Interpretation of Fig. 11: It seems that carefully chosen aerosol can bring simulated VIS006 closest to the observation. Is this conclusion correct?*

For improving the low reflectance part of the histogram adding aerosols is certainly the most important measure. However, for mid to high reflectances they are not that helpful. They have some influence on the maximum reflectance of the clouds, but the latter depends on many other factors.

*L. 398: "ice habit is thus not likely to cause large uncertainties..." This is only true because your scenes have a very high low-level cloud cover and semi-transparent cirrus overlays lower clouds, right?*

We agree on that and made this clear in the text.

*L. 403: "simulation" -> "simulations"*
We changed accordingly.

*L. 411 & l. 415: "pixels" I find "pixel" inappropriate for model data.*

We would agree if we compare in model space. However, the evaluation is carried out in observation space and we think pixel is also appropriate for synthetic images.

*L. 420: "experiment VI" -> do you rather mean VII?*
You are right. The text was revised accordingly.

*L. 424 / 425: "cloud top height is an important additional parameter" I guess you mean in addition to cloud-optical thickness? Please make this clear!*

We agree and revised the text to make this clear.

*Fig. 12:*

*\* It is hard to see the differences here. The plot are dominated by the extremes. Please trim the range of the PDFs e.g. within (-2, 2).*

*\* Panel a & b should have the same size.*

*\* y labels should be eps_n,d consistent with Sect. 2.5*

We modified Fig. 12 accordingly.

*L. 450: "to eliminate the errors of the reference run" + "in the IR108 channel"*
We added "in the IR108 channel "

*L: 485: "solar satellite observations are novel for model evaluation". This might be true for RTTOV, but not for other forward operator methods. Please be more specific and discuss, if applicable, already existing advancements by others (e.g. in CRTM).*

You are right. We meant operational applicable forward operators for model evaluation and data assimilation. A clarification has been added to the paper and we removed the word "novel".

*L. 490: "well-suited to improve model cloud parameterisations for better PW power production forecasts" This statement could be better supported by your analysis. To my feeling, you can show that better VIS / IR108 forecasts ultimately lead to improvements in GHI predictions*

We have removed this paragraph from the conclusion.

---

## Author Comment (AC2)

**Response to Referee Comment (RC2) on**

Understanding the model representation of clouds based on visible and infrared satellite observations (https://doi.org/10.5194/acp-2021-5)

We appreciate the comments of the reviewer and have revised the manuscript to address all remarks. Our response is provided in black and the review in blue.

*The manuscript presents satellite and model comparisons from 2 days during a 30-day ICON-D2 hindcast to motivate the use of visible and infrared analysis in tandem when assessing model clouds. Then statistics from the full 30 days are shown to illustrate systematic model deficiencies. An attempt is made to understand the source of these deficiencies by focusing on cloud parameters and parameterizations within ICON. Tweaks to these schemes are used to motivate possible ways to improve ICON.*

*There is a lot of back and forth in the study design between weather models, radiative transfer calculations, and satellite observations. This does not seem to be atypical for the satellite community, but for those of us on the cloud process/modeling side who would seem to benefit most from this study, this back and forth presents an opportunity for confusion. My biggest concern with the manuscript is not the methods, per se, but the logic of their presentation. I think the overall experimental design needs to be made much clearer. Why are the steps taken the right ones and taken in the right order? I had to sketch out sequencing of the study for myself after reading the manuscript a second time to make sense of things. Even then, some aspects of the manuscript felt out of place.*

We revised the abstract, introduction and various other parts to make the story/approach of the paper clearer.

Major Concerns

1. *The abstract contains lots of ambiguous sentences that simply can't stand on their own. For example, the second to last sentence means something very specific to the authors (and to the reader after reading the manuscript) but seems very unclear to the uninitiated. The same could be said of the final sentence and many others.*

The last two sentences were removed (together with most other sentences referring to surface radiation, see response to point 3 below) and the rest of the abstract was revised.

2. *The paragraph starting on L65 seems very important, but has similar issues to the abstract. This is a somewhat roundabout study which focuses on a number of different things, so I think this paragraph which is intended to describe the logic of the methodology deserves to be better. I would start by reiterating the goal of the study (which I infer to be): "the meteorologically forecasting relevant quantities for PV generation will rely on assimilating clouds well and on accurate cloud*

> simulation. This study is therefore aimed at improving our general representation of clouds in models by assessing current model performance relative to satellite observations. Etc"

Thank you for your suggestion, we understand your concern. We have rephrased the introduction, including this paragraph.

> 3. *Section 3.3 seems unnecessary. Maybe I'm missing something important, but the result of this section seems logical and the figure unsurprising.*

We understand that this part should not be in Section 3. However, we still think that for some readers, e.g. those with a data assimilation background that were so far only concerned with remote sensing observations in the infrared and microwave part of the spectrum, the difference in the correlation could be valuable information. Therefore we have removed the model equivalents from the plot and moved it to Sect. 2.2. There it fits well as it shows, based on observations, the consequences of the idealised plots for the cloud signals in infrared and visible channels.

> 4. *The exact logic of section 4.1 needs to be explained. It's not clear precisely how I should interpret this figure in general. For example, if one of your test cases exactly recreated the OBS but REF didn't, it's not exactly clear to me what the conclusion would be. What if REF and REF-Grid were exactly the same? Should this analysis be used to draw conclusions about the success or failure of ICON or of the forward model? I don't need answers to these questions, exactly, but rather am trying to illustrate my lack of understanding of the logic of this section.*

This section is intended to show the relative contributions of different model clouds (different phases, subgrid or grid-scale) to the reflectance distribution and to identify the main suspects responsible for deviations between the observed and modelled distribution. Fig. 9 allows for comparing the contribution of different cloud types (e.g. REF - REF-grid for the subgrid clouds) to the deviation between OBS and REF. If the contribution of a certain cloud type is much smaller than the deviation it is unlikely that tuning (i.e. slightly changing) these clouds in the model or making changes related to them in the forward operator will be an effective way to reduce the deviations. If these changes should be made in the operator or the model is the topic of sections 4.2 and 4.3.

The most relevant information that can be gained from Fig. 9 is probably that subgrid clouds are important, as REF and REF-grid are really different.

We added several sentences to 4.1. to explain the intention of Fig. 9.

> 5. *I am left wondering how sensitive the conclusions are to the cloud morphology of summer over northern Europe. Presumably column precipitate mass is mostly liquid during these months which leads you to the conclusions that liquid is ultimately important for (if nothing else) solar reflectance. Do you feel your results are generally applicable in the context of a weather model that may need to simulate lots of different cloud states over the course of a year?*

The general conclusion of this paper is that visible channels can provide important additional information for model evaluation as they saturate later than infrared channels. For the specific (and relevant) situation investigated here it is the water content of liquid clouds that can be constrained by the visible reflectance. For situations in which ice clouds are more predominant the late saturation should still be helpful for constraining the full (liquid plus frozen) water content, but it may be more problematic to attribute deficiencies in the full content to problems with water or ice clouds. In such cases it may be helpful to additionally consider the $1.6\mu$m channel available on many satellite instruments, which allows for distinguishing between water and ice clouds.

These remarks have been added to the results section.

**Minor Concerns**

*L96: How did you determine what is physically plausible?*

As explained in the text we target parameters whose values are unknown, i.e. not constrained by some physical law/observation. Previous to this study we talked to the developers of the different parameterizations of DWD about the plausibility of the values used for these experiments. We have individually discussed in section 2.1 which parameters have been changed and the effect on the parameterizations.

*L100: These seem arbitrarily chosen. How were these chosen before the study or were they chosen as a result of initial data analysis?*

The experiments target the parameterizations that directly influence clouds: microphysics, convective parameterization, subgrid parameterization and cloud number concentration. Some of them were chosen based on initial results.

*L126: Similarly, why these seven (especially for VI and VII)?*

See comment above; we chose simulation VI, because we found way too many ice clouds and we were not sure if the signal was caused by subgrid or grid scale clouds. Simulation VII was motivated because the two-moment scheme reflected too much radiation, and therefore we reduced the amount of subgrid clouds.

*L176-L191: You mean the effective radii calculated by the ICON radiation scheme, correct? Not the geometric radii?*
Yes, you are right, we mean the effective radii. However, we use the effective radii computed in the microphysics scheme. We have added a clarification in the text.

*L196: You mean you followed the procedure of Meirink by replacing MODIS with your ICON radiances? Why is this a necessary step? Without it, might you have usefully inferred a model bias?*

You are right. Basically, we remove a systematic bias, where the model produces too many very thick clouds. We can do that because the correct calibration of SEVIRI VIS006 solar reflectances is unknown and calibration of visible radiances is challenging.

Meirink et al. (2018) showed that SEVIRI reflectances are underestimated by 8% compared to MODIS observations. MODIS observations should be more accurate.

*L242 and L273: Use of the word "exemplarily" feels a little out of place.*

We removed "exemplarily" in L242 and L273 accordingly from the text

*Fig. 6: Do "difference plots" help to highlight anything that isn't obvious simply by showing observation and simulation results side by side?*

We also looked at difference-plots, but we think they do not reveal any important additional information in this case. As an example here is Fig. 8c) - Fig. 8d):

[Figure]

*General: It feels as though there are a lot of acronyms that have been defined but are not used very much. You may not need to define as many as you do.*

We have reduced the number of acronyms and hopefully, wherever it was possible.

*Section 4.2: I don't feel as though I have sufficient background knowledge to judge this section.*

L461: Why shouldn't they be included?
We meant for the evaluation of model clouds (changed in the manuscript) they should not be included, because the high AOD in these cases can affect the visible reflectances strongly.

---

## Referee Report (RR1)

Review of "Understanding the model representation of clouds based on visible and infrared satellite observations" by Geiss, Scheck, de Lozar, and Weissmann.

Second review by Matthew R. Igel.  Review requested 28th June 2021.  Review performed 9th July 2021.

The manuscript presents satellite and model comparisons from 2 days during a 30-day ICON-D2 hindcast to motivate the use of visible and infrared analysis in tandem when assessing model clouds. Then statistics from the full 30 days are shown to illustrate systematic model deficiencies. An attempt is made to understand the source of these deficiencies by focusing on cloud parameters and parameterizations within ICON. Tweaks to these schemes are used to motivate possible ways to improve ICON.

The authors effectively argue that using both shortwave and longwave information together to relate models to observations is more effective than using either in isolation.  I don't think this is a surprising conclusion to draw, but it is certainly worth making.  The combination of 2 snapshot case studies with statistics from 30 days of simulations is well conceived.  The framework developed for thinking about how observations and models might be usefully compared is the best part of the paper, but I think could be more systematic.  The authors make some suggestions for improving models that seem logical given their results.

The authors addressed my major concerns from round 1.

I suppose the basic question that remains unanswered by this study (to which the authors themselves allude in section 5) is whether the modifications suggested in section 4.3 from Fig. 11 can really be made.  They rely on the effects resulting from the modified-simulations being linearly additive.  I simply don't know whether they are.  Clouds are often very non-linear systems, so I think there is reason to question whether these effects would add.  But, this study does a good job of making implementable suggestions for model improvements that are derived from novel analyses from which a follow-up study can easily launch.

---

## Author Response (AR2)

*Dear Editor and Reviewer,*
*Our response to Report #2 is provided first, followed by the response to editor comments.*

**Response to Report #2 on**

Understanding the model representation of clouds based on visible and infrared satellite observations (https://doi.org/10.5194/acp-2021-5)

We appreciate the comments of the reviewer and have revised the manuscript to address all remarks. Our response is provided in black and the review in blue.

* L.100: "cloud-concentration number" -> cloud droplet number concentration
Revised accordingly: "cloud-concentration number" -> "cloud droplet number concentration"

* L.221: Please rephrase such that the -13 % difference between obs and sim can be partly interpreted as calibration offset and as model bias.
revised accordingly: "Through this, we found a model bias of -13% between observations and our reference simulation." -> "Through this, we found a deviation of -13 % between observations and our reference simulation, which can be partly attributed to a calibration offset (observation too dark) and a model bias."

* L. 230: "calibrated observations" -> term not used elsewhere
we have removed the word "calibrated"

* Fig. 10 caption: "using maximum-random instead of random ... overlap". It is exactly the other way around, right?
Thanks for spotting this! You are right. We have revised accordingly: "maximum-random instead of random" -> "random instead of maximum-random""

* L.339: "Fig 2c": reference does not exist.
Thank you. We have removed the reference and changed "total column ice content is much smaller than the water content" -> "ice water path is much smaller than the liquid water path"

**Response to Editor comments**

Understanding the model representation of clouds based on visible and infrared satellite observations (https://doi.org/10.5194/acp-2021-5)

Dear Philip Stier,
We appreciate your comments and have revised the manuscript to address all remarks. Our response is provided in black and the review in blue.
* * *
Dear authors,

Thank you very much for addressing the comments raised in the reviews in the revised version of your manuscript. This has now undergone re-review and I am pleased to inform you that your manuscript is now accepted for publication subject to (very) minor revisions.

Could you please address the few editing issues raised in reviewer report 2?
We have revised the manuscript to address all the remarks in reviewer report 2.

In addition, could you please also ensure that: all figure captions and labels are fully self-explanatory? Instead of relying on acronyms defined in the main text (e.g. VIS006, IR108), please define such terms in the captions. Captions and labels of Fig. 11 and 12 are not clear without searching the main text.

We have updated and changed (almost) all captions.

And finally, I would suggest - but not require - to change "verification" to "evaluation" as verification implies a binary true/false assessment, which is generally not possible to establish in model evaluation.
Changed accordingly: verification > evaluation

Best regards,

Philip Stier